# Beyond PD-L1—Identification of Further Potential Therapeutic Targets in Oral Cancer

**DOI:** 10.3390/cancers14071812

**Published:** 2022-04-02

**Authors:** Manuel Weber, Rainer Lutz, Manuel Olmos, Jacek Glajzer, Christoph Baran, Christopher-Philipp Nobis, Tobias Möst, Markus Eckstein, Marco Kesting, Jutta Ries

**Affiliations:** 1Department of Oral and Maxillofacial Surgery, Friedrich-Alexander University Erlangen-Nürnberg (FAU), 91054 Erlangen, Germany; manuel.weber@uk-erlangen.de (M.W.); rainer.lutz@uk-erlangen.de (R.L.); manuel.olmos@uk-erlangen.de (M.O.); jacek.glajzer@uk-erlangen.de (J.G.); christoph.baran@uk-erlangen.de (C.B.); christopher-philipp.nobis@uk-erlangen.de (C.-P.N.); tobias.moest@uk-erlangen.de (T.M.); marco.kesting@uk-erlangen.de (M.K.); 2Comprehensive Cancer Center Erlangen-EMN (CCC ER-EMN), 91054 Erlangen, Germany; 3Institute of Pathology, Friedrich-Alexander University Erlangen-Nürnberg (FAU), 91054 Erlangen, Germany; markus.eckstein@uk-erlangen.de

**Keywords:** immune checkpoints, OSCC, immunotherapy, head and neck cancer, HNSCC, macrophage, MDSC, T-cell

## Abstract

**Simple Summary:**

Tumor immunotherapy is rapidly evolving and approved for the treatment of advanced OSCC cases. In addition, the currently observed shift in the use of checkpoint inhibitors from palliative to neoadjuvant treatment may improve survival. However, not all patients respond to currently applied immune checkpoint inhibitors. Therefore, further immune targets for therapeutic approaches are urgently needed. However, there are limited data on immune checkpoint expression in OSCC. This study aimed to perform a comparative analysis of a large number of immune modulators in OSCC compared with healthy controls by NanoString mRNA analysis in order to identify possible targets for therapeutic applications. We were able to ascertain several cellular markers, checkpoints and their correlation, as well as their association with histomorphological parameters. Hence, the study contributes to the understanding of immune escape in OSCC and reveals potential targets for immunotherapy of oral cancer.

**Abstract:**

Background: The involvement of immune cell infiltration and immune regulation in the progression of oral squamous cell carcinoma (OSCC) is shown. Anti-PD-1 therapy is approved for the treatment of advanced OSCC cases, but not all patients respond to immune checkpoint inhibitors. Hence, further targets for therapeutic approaches are needed. The number of identified cellular receptors with immune checkpoint function is constantly increasing. This study aimed to perform a comparative analysis of a large number of immune checkpoints in OSCC in order to identify possible targets for therapeutic application. Materials and Methods: A NanoString mRNA analysis was performed to assess the expression levels of 21 immune regulatory checkpoint molecules in OSCC tissue (*n* = 98) and healthy oral mucosa (NOM; *n* = 41). The expression rates were compared between the two groups, and their association with prognostic parameters was determined. Additionally, relevant correlations between the expression levels of different checkpoints were examined. Results: In OSCC tissue, significantly increased expression of CD115, CD163, CD68, CD86, CD96, GITRL, CD28 and PD-L1 was detected. Additionally, a marginally significant increase in CD8 expression was observed. BTLA and PD-1 levels were substantially increased, but the differential expression was not statistically significant. The expression of CD137L was significantly downregulated in OSCC compared to NOM. Correlations between immune checkpoint expression levels were demonstrated, and some occurred specifically in OSCC tissue. Conclusions: The upregulation of inhibitory receptors and ligands and the downregulation of activators could contribute to reduced effector T-cell function and could induce local immunosuppression in OSCC. Increased expression of activating actors of the immune system could be explained by the increased infiltration of myeloid cells and T-cells in OSCC tissue. The analysis contributes to the understanding of immune escape in OSCC and reveals potential targets for oral cancer immunotherapy.

## 1. Introduction

Oral squamous cell carcinoma (OSCC) is the most common tumor of the oral cavity, accounting for more than 90% [1]. In 2020, there were 377,713 cases of lip and oral cavity carcinoma worldwide, making the disease the fifth most common carcinoma. Of the patients with the disease, 177,757 died [2]. 

Surgical resection with adjuvant radio(chemo)therapy in high-risk cases represents an effective therapy for oral squamous cell carcinoma (OSCC). Additionally, the current surgical approach has been further improved by the use of innovative methods such as robotic surgery, which allow not only an R0 approach but also a rescue operation that allows de-intensified adjuvant chemoradiotherapy [3]. However, besides initial R0 resection, recurrences often occur, requiring salvage surgery, re-irradiation or palliative systemic treatment with unsatisfying overall survival and quality of life [4].

The introduction of anti-PD-1 immunotherapy allowed long-term survival in a small proportion of patients [5]. Today, anti-PD-1 immunotherapy is shifting from the palliative setting to earlier disease stages up to neoadjuvant immunotherapy, which additionally increases the response rate [6,7]. However, more than half of patients do not respond even to modern neoadjuvant immunotherapy protocols. This primary resistance to ICIs could be due to an abnormal composition of the gut microbiome. Antibiotics have been shown to interfere with the clinical benefit of ICIs in patients with advanced cancer. Metagenomics of patients’ stool samples at diagnosis revealed an association between clinical response to ICIs and the relative abundance of Akkermansia muciniphila. Oral supplementation with A. muciniphila after fecal microbiota transplantation with feces from patients who did not respond to treatment increased the efficacy of PD-1 blockade in mouse tumors. Therefore, such a therapeutic approach could be useful for tumor patients [8]. Another strategy attempting to further improve response rates is to combine anti-PD-1 therapy with other immunotherapeutic drugs, such as the combination of PD-1- with cytotoxic T-lymphocyte-associated antigen 4 (CTLA-4) inhibitors [9] or the addition of macrophage-modulating drugs [10]. This underlines that further targets are needed, and a combination of immune modulators might be a promising approach. However, even though the number of identified cellular receptors with immune checkpoint function is constantly increasing, a comparative analysis of a large number of immune checkpoints in OSCC is lacking.

Checkpoint inhibitor-based immunotherapy for solid tumors first became available with CTLA-4 inhibitors, followed by PD-1 inhibitors [11,12]. However, a variety of inhibitory and activating immune checkpoint signaling pathways that finely regulate the immune response have since been described [11,13]. 

Many of immune modulators are suitable pharmacological targets, such as lymphocyte activation gene-3 (LAG-3) [14] or T-cell immunoglobulin 3 (Tim3) [14,15]. The mechanism of action of immune checkpoint molecules beyond PD-1 and CTLA-4 is incompletely understood [13]. It has been proposed that the T-cell-based LAG-3 receptor interacts with MHC class II molecules and leads to decreased T-cell cytokine production, decreased proliferation of effector T-cells and the induction of regulatory T-cells (T_reg_), which causes tumor immune escape [13]. Besides MHC II, fibrinogen-like protein 1 (FGL1) is a potential binding partner of LAG-3 [16]. Tim3 is an immune-suppressive receptor on T-cells that is activated by several ligands, including the lectin galectin-9 (Gal9) [17,18]. In the context of cancer, Tim3 is a marker for highly exhausted dysfunctional T-cells [18]. Besides LAG-3 and Tim3, there are many other potentially therapeutically relevant immune checkpoints. T-cell immunoglobulin and ITIM domain (TIGIT) is a T-cell- and NK-cell-based inhibitory receptor [19]. The main TIGIT ligand, CD155, is expressed on tumor-infiltrating myeloid cells, macrophages and also tumor cells [20]. CD96 is another T- and NK-cell-based checkpoint receptor that can bind to CD155. While an inhibitory effect of CD96 is assumed based on mouse models, its function in human tissues still needs to be clarified [19]. The B- and T-lymphocyte attenuator receptor (BTLA) is an inhibitory receptor of B- and T-cells. BTLA expression indicates terminally exhausted lymphocytes and was shown to be correlated with lung cancer progression [21]. CD137 is an activating checkpoint receptor of T-cells and NK cells but also of antigen-presenting cells [22]. The corresponding ligand CD137L is expressed on antigen-presenting cells [23]. There are early ongoing trials with activating CD137 antibodies for cancer immunotherapy [22,23]. Glucocorticoid-induced tumor necrosis factor receptor (GITR) is an activating receptor of T-lymphocytes. Its ligand GITRL is expressed on antigen-presenting cells and macrophages. GITR signaling leads to both activation of T-cells and inhibition of T_reg_ cells [24,25]. CD80 and CD86 are among the most important co-activating ligands expressed by antigen-presenting cells besides MHCI/II during T-cell activation [26,27]. CD80/86 bind to CD28 on T-cells transducing the main activating signal. However, CD80/86 can also bind to the T-cell-based CTLA-4 receptor, which mediates immune suppression [26,27]. An overview of the checkpoint molecules and cellular parameters analyzed in this study is presented in Figure 1.

Besides immune checkpoints, the cellular infiltrate in human malignancies, including HNSCC, is highly relevant for tumor progression, response to therapies and survival [28]. In this regard, T-cells, macrophages and myeloid-derived suppressor cells (MDSCs) are of special interest. CD8-positive T-cells are the main effector cells of anti-tumor immunity [29]. Various tumor entities (including NSCLC, melanoma and others) with high CD8 infiltration were shown to have a better response to immunotherapy [29]. However, a recent meta-analysis showed no association between CD8 infiltration and survival in oral cancer [30]. The infiltration of tumor-associated CD68-positive macrophages and, in particular, tumor-promoting, M2-polarized, CD163-expressing macrophages was shown to be associated with oral cancer initiation and tumor progression [31,32,33]. Nevertheless, although cancer-associated fibroblasts (CAFs) are important stromal cells, the characterization of their heterogeneity is far from complete. It has recently been shown that upregulated CD68+ fibroblasts are involved in tumor initiation, but the subset of CAFs with low CD68 expression in OSCC is conducive to the recruitment of regulatory T-cells (Treg) in the tumor microenvironment and contributes to a poor prognosis of OSCC patients [34]. CD115, also known as colony-stimulating factor 1 receptor (CSF1R), is a surface receptor expressed on tumor-associated macrophages [35]. In addition, CD115 is expressed by a heterogeneous myeloid cell population with immune-suppressive properties, designated myeloid-derived suppressor cells (MDSCs) [35]. IL-10 is a major immune-suppressive cytokine secreted by MDSCs [36], M2 macrophages [31] and T_reg_ cells [37]. Figure 1 displays an overview of the analyzed cell populations and cytokines in the current study.

Even though the number of identified cellular receptors with immune checkpoint function is constantly increasing, in OSCC, the expression of many of the currently described immune checkpoints has not been comprehensively analyzed so far. Therefore, the aim of this study was to provide a comparative analysis of immune checkpoints and immune cell markers as a basis for further targeted investigation of checkpoints and immune evasion in OSCC. Hence, 16 immunologically relevant checkpoints, together with cellular markers and cytokines, were analyzed at the mRNA level.

## 2. Materials and Methods 

### 2.1. Patients and Sample Collection

Formalin-fixed paraffin-embedded tissue specimens from 98 patients suffering from OSCC (OSCC group) and 41 healthy volunteers (control group, normal oral mucosa, NOM) were included. The study was approved by the Ethics Committee of the University of Erlangen-Nuremberg, Erlangen, Germany (approval number: 3962), and patients’ written informed consent was obtained. Age and gender of all study participants were collected. Patients were only included if it was an initial diagnosis of oral malignancy and no previous treatment had been received. The selection of healthy subjects was based on the absence of tumor disease, general disease, and acute or chronic inflammatory conditions of the normal oral mucosa. TNM classification, staging and evaluation of differentiation (G1 = well, G2 = moderately, G3 = poorly differentiated) of malignant tissues were performed by two independent pathologists according to the guidelines of the World Health Organization and the International Union Against Cancer [38]. Additionally, tumors were grouped into small (T1 and T2) and large (T3 and T4) malignancies, early (stages I and II) and late (stages III and IV) clinical stages and N0 and N+ (indicating a negative or positive lymph node status).

### 2.2. Sampling of Tumor and Normal Oral Mucosal Specimens

Specimens from healthy volunteers were taken during minor intraoral surgery to avoid making additional incisions, fixed in 4% formalin and subsequently embedded in paraffin. Paraffin-embedded tumor tissues samples were obtained from the Institute of Pathology of Friedrich-Alexander University Erlangen-Nürnberg. All tumor samples used in this analysis were cut, H&E-stained and analyzed using a microscope in order to ensure a malignant tissue content of at least 80%. 

### 2.3. RNA Isolation and Expression Analysis

The miRNeasy Mini Kit (Qiagen, Hilden, Germany, cat. no. 217504) was used for whole RNA isolation from tissue samples. RNA concentration was measured with a NanoDrop spectrometer (PeqLab, Erlangen, Germany). The RNA samples were stored at −80 °C until expression analysis.

For expression analysis of the immune modulators, the nCounter™ System (NanoString Technologies^®^, Inc., Seattle, WA, USA) was used. We performed the nCounter™ assay (panel was made on demand for customer solutions; NanoString Technologies^®^, Inc., Seattle, WA, USA) using 400–500 ng of total RNA per replicate. In order to improve the precision of the measurements, each assay was performed in duplicate. The nCounter™ gene probes (NanoString Technologies^®^, Inc., Seattle, WA, USA) include 21 immune-modulatory genes and 3 housekeeping genes (GAPDH, polymerase II, PPL19) for normalization. In addition, the nCounter™ Analysis System (NanoString Technologies^®^, Inc., Seattle, WA, USA) routinely includes spike-in controls for calibration and quality control. Transcripts for the nCounter™ assay and nCounter PlexSet Design and Isoform Coverage are listed in Table 1 (panel was made on demand; NanoString Technologies^®^, Inc., Seattle, WA, USA). The data were evaluated by the NanoString analysis software nSolver^®^ (NanoString Technologies^®^, Inc., Seattle, WA, USA).

### 2.4. Statistical Analysis

For statistical analyses, the average of the duplicates of the determined counts was applied. The relative gene expression was calculated as the ratio of mean counts of the groups and is reported as fold change. A value greater than 2 is considered to be relevantly increased. For statistical evaluation, the software package SPSS 23 (SPSS Inc., Chicago, IL, USA) was used. In order to determine significant differential expression between the groups, Kruskal–Wallis test and Mann–Whitney U test (MWU) were applied. A *p*-value ≤ 0.05 was considered to be statistically significant. In order to visualize the data, box-whisker plots were utilized. Correlation analysis was performed using the Spearman correlation test. Spearman correlation values and two-sided adjusted *p*-values are reported.

## 3. Results

### 3.1. Demographic and Clinico-Histopathological Characteristics of the Study Participants

The demographical, clinical and histopathological features of all study participants are summarized in Table 2. The two groups of the collective (NOM and OSCC) were gender-matched (*p* = 0.27). However, the mean age of the healthy control group (NOM) was significantly lower than that of the OSCC group (*p* = 0.01). None of the healthy volunteers had remarkable oral mucosal changes such as inflammation, hyperplasia or dysplasia.

Out of the 98 OSCC cases, 58 were classified as small (T1/T2: 59.2%), and 39 were classified as large (T3/T4: 39.8%) malignancies. In 51 cases, the lymph nodes were not affected (N0: 52%). Forty-seven cases showed lymph node metastases (N+: 48%). Of the OSCC tissues analyzed, 7 were well differentiated (G1: 7.1%), 54 were moderately differentiated (G2: 55.1%) and 35 were poorly differentiated (G3:35.7%). The clinical stage was classified as early in 33 OSCC patients (33.7%) and late in 64 patients (65.3%). During follow-up, 25 patients suffered from recurrence (25.5%), while 62 patients (63.3%) did not show recurrent disease. For 11% of the patients, no data were available. The mean disease-free survival (DFS), which describes the period after successful treatment until the occurrence of relapse, amounted to 12.7 months. 

### 3.2. Comparison of Expression Rates of Immune Modulators between OSCC and NOM

Statistical analysis revealed that out of the 21 investigated immune modulators, 9 were significantly differentially expressed in OSCC compared to NOM. Thus, the cell marker for myeloid cells and MSDCs, CD115 (*p* = 0.0001, FC = 3.02), and the macrophage markers CD68 (*p* = 0.0001, FC = 4.04) and CD163 (*p* = 0.009, FC = 1.8) were significantly increased in the OSCC group. The levels of the activating checkpoint receptor CD86 (*p* = 0.03, FC = 1.33) and the activating ligand CD28 (*p* = 0.006, FC = 1.75) were significantly increased, but the altered expression levels were less than two-fold. The inhibitory receptor CD96 (*p* = 0.02, FC = 1.79) and the inhibitory ligands PD-L1 (*p* = 0.002, FC = 4.07) and GITRL (*p* = 0.04, FC = 1.58) were significantly overexpressed in malignancy, but the change amounted to less than two-fold for CD96 and GITRL. Significantly decreased expression was detected for CD137L (*p* = 0.02, FC = −1.5). The cytotoxic T-cell marker CD8 and the inhibitory receptors CTLA-4, BTLA and PD-1 showed at least two-fold increased expression levels in OSCC compared with NOM (FC_CD8_ = 2.76, FC_CTLA-4_ = 1.99, FC_BTLA_ = 2.44, FC_PD-1_ = 2.2). The results are summarized in Table 3 and visualized in Figure 2. 

### 3.3. Association of Differential Expression Patterns with Clinico-Histopathological Parameters

Regarding expression changes with respect to grouped tumor size, weak statistically significant downregulation of CD115 (*p* = 0.05; FC = −1.6), CD68 (*p* = 0.04, FC = −1.3) and CD137L (*p* = 0.04, FC = −1.7) was seen in larger lesions (T3/T4). PD-L1 (*p* = 0.001, FC = −5.7) and IL10 (*p* = 0.009, FC = −1.3) expression was significantly lower in the T3/T4 tumor group (Table 4).

Statistical analysis revealed that only CD68 (*p* = 0.05, FC = 1.56) was marginally relevantly upregulated in tumor tissues of patients suffering from lymph node metastases (Table 4).

A statistical correlation between the UICC stage and the differential expression of any immune-modulatory protein could not be demonstrated.

In tumor tissues of patients suffering from recurrent disease in the follow-up period, CD80 was significantly downregulated (*p* = 0.02, FC = −2.2). The fold change in altered CTLA-4 expression regarding the UICC stage was −2.1, but the significance of this downregulation could not be proven (*p* = 0.07; Table 4). 

The Kruskal–Wallis test revealed that differentiation (grading) was associated with differential expression of CD115 (*p* = 0.02), CD68 (*p* = 0.01), CD163 (*p* = 0.001) and PD-L1 (*p* = 0.03) (Table 4). Further analysis comparing the expression levels between groups of different degrees of differentiation (G1, G2, G3) showed that with the exception of PD-L1, the expression of immune modulators decreased with higher dedifferentiation. The expression of the immune checkpoint PD-L1 was slightly but significantly increased in G2 versus G1 tumors, probably due to outliers and extreme outliers (Table 5, Figure 3). The alterations in expression were significant between well (G1) and moderately or poorly differentiated tissues (G2 and G3), respectively. No statistically significant decrease was seen when comparing the G2 and G3 groups. 

### 3.4. Correlations between Receptors, Matching Ligands and Corresponding Immune Cells

#### 3.4.1. PD-1–PD-L1/PDL2 Axis

No correlation was seen between PD-1 expression and PD-L1 or PD-L2. The expression levels of PD-L1 and PD-L2 were moderately correlated (Spearman’s ρ = 0.26, *p* < 0.002; Appendix A).

#### 3.4.2. CD155, CD96 and TIGIT

No correlation was seen when comparing CD155 expression to TIGIT. The expression levels of CD155 and CD96 were moderately correlated (Spearman’s ρ = 0.39, *p* < 0.001). A weak correlation was proven between the expression of CD96 and TIGIT (Spearman’s ρ = 0.24, *p* = 0.005). If only the NOM group was taken into account, the correlation was lost (Spearman’s ρ = −0.265, *p* = 0.094). On the other hand, TIGIT and CD96 expression were strongly correlated (Spearman’s ρ = 0.43, *p* = 0.0001; Appendix A). 

#### 3.4.3. CD68, CD163 and the MDSC Marker CD115

The expression levels of the MSDC marker CD115 and macrophage markers, as well as those of CD68 and CD163, were strongly positively correlated (all cases: *p* < 0.001, Spearman’s ρ summarized in Appendix A). These results were also seen in the OSCC and NOM groups. However, the association of CD68 expression with CD115 and CD163, although still significant, decreased from a strong correlation (ρ = 0.594 and 0.773) to a weak correlation (CD115: ρ = 0.211) or, in the case of CD163 (ρ = 0.404), to a moderate correlation in the NOM group compared with the OSSC group (Appendix A).

#### 3.4.4. PD-L1 Expression and Macrophage Infiltration (CD68) and Polarization (CD163)

PD-L1 and CD68 expression were strongly correlated in the OSSC group (Spearman’s ρ = 0.527, *p* < 0.001) but not in the NOM group (Spearman’s ρ = 0.249, *p* = 0.116). The correlation of CD163 with PD-L1 expression was significant in all groups (all samples, OSCC, NOM). However, the relationship was stronger in the OSCC group (ρ = 0.739) compared to NOM (ρ = 0.483: Table 6, Figure 4). 

PD-L2 expression did not correlate with the expression of CD68 or CD163 in the OSCC and NOM groups. A very weak correlation could be seen if all samples were considered (Table 6).

#### 3.4.5. CD28/CTLA-4–CD80/86 and the Cytotoxic T-Cell Marker CD8

Correlation analysis of the CD28/CTLA-4–CD80/86 signaling pathway was initially performed for OSCC and NOM tissues combined (*n* = 140). No correlation between CD86 and CTLA-4 expression was detected. The expression of CD28 was weakly correlated with CD86 (Spearman’s ρ = 0.289, *p* < 0.001) and moderately correlated with CD80 (Spearman’s ρ = 0.393, *p* < 0.001) and CTLA-4 (Spearman’s ρ = 0.405, *p* < 0.001). A weak correlation was proven between the expression of CD80 and CD86 (Spearman’s ρ = 0.196, *p* = 0.02). Moderate correlations of expression were seen between CTLA-4 and CD80 (Spearman’s ρ = 0.291, *p* < 0.001; Table 7A). 

When considering only the OSCC group (*n* = 98), the weak correlation between CD86 and CD28 or CD80 was not observed. The other positive correlations between the members of the CD28/CTLA-4–CD80/86 axis seen in the analysis of all samples were stronger and changed from moderate to strong (Spearman’s ρ = 0.535, Spearman’s ρ = 0.514) except for CD80/CTLA-4 (Spearman’s ρ = 0.398; Table 7B). The weak correlation between CD80 and CD86 or CD28 could not be confirmed in OSCC samples. All results are shown in scatter plots (Figure 5B).

In the NOM group (*n* = 42), no correlations between the expression of receptors and ligands could be demonstrated. This could indicate that all correlations were only relevant in OSCC patients. 

An exception is the correlation between CD80 and CD86, which showed a strong relationship in the NOM group (Spearman’s ρ = 0.623, *p* = 0.001; Table 7C) but not in the OSCC group. All results are visualized in Figure 5.

The expression of CD8 as a cytotoxic T-cell marker was moderately associated with CD28 and CTLA-4 expression when assessing all cases (*n* = 139). The relationship was positive for CD28 (Spearman’s ρ = 0.269, *p* < 0.001) and negative for CTLA-4 (Spearman’s ρ = −0.227, *p* < 0.007; Table 7A). These results were also observed for the OSCC group (Table 7B, Figure 5B). However, if only the NOM samples were taken into consideration, no significant correlation was found (Table 7C, Figure 5).

#### 3.4.6. CD137 and Ligand CD137L

The expression of the receptor CD137 and that of its ligand CD137L were strongly positively correlated (all cases: Spearman’s ρ = 0.42, *p* < 0.001; Appendix A)

### 3.5. Alterations in the Expression Ratio of the Antagonistic Immune Modulators CTLA-4/CD28 Related to Diagnosis and Prognostic Parameters

There was no statistical correlation between the ratio of CTLA-4 to CD28 expression and diagnosis (*p* = 2.2). No significant association with tumor size (*p* = 0.19), lymph node status (*p* = 0.63), grading (*p* = 0.23) or the clinical UICC stage (*p* = 0.92) was detected. The ratio was significantly higher (two-fold) in tumor tissue from patients who experienced recurrence during follow-up (*p* = 0.036; FC = 2.2; ratio NOM 4.85; ratio OSCC 10.49; Figure 6). 

### 3.6. Expression of Immune-Modulatory Proteins in Relation to Cellular Markers in OSCC and NOM

#### 3.6.1. Immune-Modulatory Gene Expression in Relation to the Cytotoxic T-Cell Marker (CD8)

In order to determine the infiltration of the specimens with cytotoxic T-cells, the expression of CD8 was analyzed. The ratios of the immune checkpoints CTLA-4, LAG-3, PD-1, TIM3, TIGIT, CD96 and BTLA as well as the immune stimulatory ligands CD28, CD137 and CD80 to CD8 were calculated and compared between the groups. Significant differences were seen in the ratios of LAG-3, TIGIT and CD137 between NOM and OSCC. Elevated levels were found for CD137, whereas TIGIT was less expressed relative to CD8 expression. The expression of PD-1 and BTLA vs. CD8 was strong but not significantly increased in OSCC, as shown by a ratio of over 4. The results are summarized in Table 8a. 

#### 3.6.2. Immune-Modulatory Gene Expression in Relation to the MDSC Marker CD115

CD115 is a cell marker for MDSC cells and myeloid cells, from which a variety of immune cells, such as macrophages, granulocytes and dendritic cells, develop. Hence, the expression of this marker was analyzed to assess the infiltration of the specimens by myeloid cells and MDSCs. Then, the ratio between different cell markers and CD115 was determined. The values of the OSCC group were compared to NOM. Significant changes were seen for all ratios except for CD68/CD115. However, only the PD-L1-to-CD115 ratio was strongly increased in OSSC (3.37) compared to NOM (1.06), whereas the CD137L-to-CD115 and CD155-to-CD115 ratios were higher in the NOM group compared to the OSCC group (Table 8c).

#### 3.6.3. Immune-Modulatory Gene Expression in Relation to Macrophage Infiltration

The infiltration of the tissues by macrophages was defined by analyzing the expression of the general marker CD68. Then, the ratios between different cell markers (CD163, CD80, CD86, GIRTL, CD137L, CD155, PD-L1 and PD-L2) and CD68 were determined. The values in the OSCC group were compared to those in NOM. Significant changes were seen for all ratios except for the GIRTL- and PD-L1-to-CD68 ratios. However, the PD-L1-to-CD68 ratio was strongly increased in OSSC (4.30) compared to NOM (0.73). The results are summarized in Table 8c.

#### 3.6.4. Immune-Modulatory Gene Expression in Relation to M2 Macrophage Polarization Characterized by CD163 Expression

The polarization of the macrophage population toward M2 was characterized by the expression of CD163. Then, the ratios between different cell markers (CD80, CD86, GIRTL, CD137L, CD155, PD-L1 and PD-L2) and CD163 were determined. The values in the OSCC group were compared to those in NOM. Significant changes were seen in CD137L- and CD155-to-CD163 ratios. However, the PD-L1-to-CD68 ratio was strongly but not significantly increased in OSSC (4.32) compared to NOM (0.74). The results are summarized in Table 8d.

## 4. Discussion

### 4.1. Immune Cells and Cytokines

The cytotoxic T-cell marker CD8 was upregulated more than two-fold in OSCC compared to NOM tissues; however, it failed to reach the statistical significance level. This indicates a dysregulation of the immune cell infiltrate in OSCC that is associated with malignant growth. No association between CD8 expression and prognostic markers (TNM, grading, recurrence) was demonstrated, with the exception of the UICC. CD8-positive T-cells are relevant for tumor immunology and immune therapy response [39]. The presence of CD8+ tumor-infiltrating lymphocytes is associated with better survival in many different carcinomas, including HNSCC. However, in OSCC, their prognostic value remains unclear [30,40], and it is most likely that the location of infiltration in tumor tissue influences the clinical outcome of patients [40,41]. Although CD8 infiltration alone does not seem to be a predictor of OSCC survival [30], an association of the CD8/Foxp3 ratio with survival was shown [42]. In patients receiving anti-PD-1 immunotherapy, a high degree of CD8 infiltration might also improve treatment response [39]. Interestingly, OSCC patients with a late UICC stage showed significantly reduced expression of CD8 in the current analysis, indicating the possible impairment of the T-cell response in advanced oral cancer. 

The expression of the myeloid cell marker CD115, as well as that of the macrophage markers CD68 and CD163, was significantly increased in oral cancer samples compared to healthy oral mucosa. Immunohistochemical analyses previously showed that increased macrophage infiltration in OSCC is associated with parameters of tumor malignancy and tumor progression [33,43]. In addition, an increase in macrophage infiltration and a shift towards CD163-positive, M2-polarized macrophages in oral leukoplakia were associated with malignant transformation during a 5-year follow-up [32]. In a recent phase II study on neoadjuvant immunotherapy in OSCC using a PD-1 inhibitor in combination with the tyrosine kinase inhibitor Sitravatinib, a histological response was detected in 90% of the cases [10]. Sitravatinib is believed to re-polarize tumor-promoting M2 macrophages towards anti-tumoral M1 macrophages [10]. This underlines the potential of macrophage modulation in immunotherapy of OSCC.

MDSCs are a heterogeneous cell population of immature myeloid cells with immune regulatory properties that are not yet well characterized. These cells are important sources of immunosuppressive cytokines such as IL-10 [36]. CD115 is expressed on MDSCs as well as on macrophages and other myeloid precursor cells [20]. This is supported by our results, as the expression rates of CD115 were strongly correlated with the macrophage markers CD68 and CD163 in the OSCC and NOM groups. However, CD115 also acts as a cellular receptor for the cytokine macrophage colony-stimulating factor (MCSF). Activation of CD115 can shift macrophage polarization towards tumor-promoting M2 cells [35,44]. Hence, therapeutic inhibition of the CD115 receptor can re-polarize macrophages towards M1 and is investigated for its potential role in cancer immunotherapy [35,44]. The high CD115 expression detected in the current analysis indicates a potential for targeting CD115 for macrophage modulation in oral cancer.

The present study also revealed a correlation between myeloid cell marker expression and tumor grading. Interestingly, G1 OSCC showed significantly increased CD115, CD68 and CD163 expression compared to G2 and G3 cases. This finding is not consistent with previous data on protein levels that showed increased infiltration of CD68- and CD163-positive cells in G2 compared to G1 OSCC [43]. However, these analyses were performed on early-stage T1/T2, N0 OSCC biopsy samples [43]. The data show that even in cases with low histomorphologic grading, a high degree of immune cell dysregulation is present.

Antigen-presenting cells (APCs) such as macrophages are among the most relevant cells that express activating and inhibitory immune checkpoint ligands to modulate and fine-tune T-cell immune responses [13]. 

### 4.2. Activating and Inhibitory Immune Checkpoints

Immune checkpoint ligands are physiologically expressed predominantly by APCs but can also be expressed by tumor cells [13]. The corresponding checkpoint receptors are expressed by T-cells and eventually by NK cells, B-cells and further populations [13]. 

The PD-L/PD-1 signaling pathway is currently the main target for immunotherapy in OSCC. This study showed the significant upregulation of PD-L1 in OSCC, while there was no significant increase in the expression of the PD-L2 ligand. In addition, a more than two-fold increase in the expression of the PD-1 receptor was seen, but the change did not reach the level of statistical significance. Previous studies also showed increased expression of PD-L1 and PD-1 in OSCC at the mRNA [45,46] and protein levels [47]. An analog upregulation was seen for the PD-L2 ligand [48]. OSCC precursor lesions (oral leukoplakia) with malignant transformation showed significantly increased PD-L1 expression in immunohistochemistry but not in RT-PCR [49].

The current analysis also showed increased PD-L1 expression in G1 OSCC cases compared to tumors with grades G2 and G3. This is consistent with the increased expression of myeloid cell and macrophage markers in G1 cases, as these cells might be a source of PD-L1 expression. The relatively low proportion of G1 cases in the analyzed patient collective could explain this finding. However, these results also show that in cases with a low degree of histomorphologically characterized malignancy, a highly immune-suppressive microenvironment is possible.

PD-L1 and PD-L2 are expressed on macrophages, the most important antigen-presenting immune cells (APCs). Hence, in this study, the correlation of the expression of these checkpoints with macrophage infiltration and M2 polarization was evaluated. A strong correlation between PD-L1 and CD163 was seen in both the NOM and OSCC groups. Interestingly, PD-L1 and CD68 expression were strongly correlated in the OSSC group but not in the NOM group. This result was confirmed by determining the relation of the expression of PD-L1 with both CD68 and CD163. This suggests that the infiltrating macrophages, independent of their polarization if determined by CD163 expression, increasingly express PD-L1 in tumor tissue. Alternatively, one could assume that other immune cell populations or the tumor cells themselves express this checkpoint and are crucially involved in immune suppression in tumor tissues. However, this hypothesis still needs to be assessed by further studies.

The CD80/86–CD28 signaling pathway is one of the most important co-activating signals for T-cell activation [27]. However, this signaling axis is more relevant for CD4-positive helper cells compared to cytotoxic CD8 T-cells [27]. The current analysis revealed the significant upregulation of the CD86 ligand—but not the CD80 ligand—in OSCC. In addition, the activating T-cell-based CD28 receptor showed significant upregulation. These findings could indicate an increased potential for T-cell activation in OSCC tissue. 

The inhibitory checkpoint receptor CTLA-4 competes with CD28 for CD80/86 binding. However, in contrast to CD28, it transduces an inhibitory signal to effector T-cells [27]. The expression of CTLA-4 in OSCC was upregulated nearly two-fold, but statistical significance was not reached. 

It has been shown that CD80 and CD86 are differentially regulated in physiologic T-cell activation. There are data showing that CD86 is expressed in the initial phases of the immune response, and later, CD80 is upregulated [27]. When analyzing normal oral mucosa, a highly significant and strong positive correlation between CD80 and CD86 expression was seen. This correlation was lost in OSCC tumor tissue. These data indicate the potential dysregulation of CD28-mediated T-cell co-activation in OSCC. In contrast, OSCC samples showed a positive correlation between CTLA-4 and CD28 as well as CD80 and, in addition, a correlation between CD80 and CD28 that was not apparent in normal oral mucosa. This demonstrates a dysregulation of the CD80/86–CD28/CTLA-4 signaling axis in OSCC compared to healthy mucosa.

In OSCC cases with recurrence in the follow-up period, significantly decreased CD80 expression was seen. This might indicate that reduced T-cell activation is associated with disease recurrence.

Due to their competition for binding to the ligands CD80 and CD86 and their antagonistic function in the control of the immune system, one could claim that the ratio of expression levels between CTLA-4 and CD28 in the tumor shifts towards CTLA-4, which would result in a higher CTLA-4/CD28 quotient. However, this was not observed. Only patients who developed recurrence showed a significant increase in CTLA-4/CD28 ratios and, therefore, a shift towards immune suppression. Patients with an increased risk of recurrence could consequently benefit from a combined therapeutic approach addressing the shift of this quotient.

OSCC tumor samples showed significant downregulation of the activating CD137L ligand, while there was no significant difference in the expression of the CD137 receptor. In addition, CD137L expression in larger tumors (T3/T4) was significantly reduced compared to smaller OSCC. Moreover, the expression levels of the receptor CD137 and its ligand were strongly correlated. CD137 signaling predominantly activates CD8-positive T-cells, prevents them from apoptosis and enhances T-memory-cell differentiation [50]. In addition, CD137 signaling can activate NK cells and inhibit T_reg_ cells and therefore exert anti-tumoral effects in several ways. In animal models, CD137-activating antibodies showed anti-tumor activity [50]. Currently, there are a number of early human trials ongoing, especially on combinations with other checkpoint inhibitors in several malignancies, including breast cancer, colon cancer and other advanced malignancies [22,23,50]. The data from the current study underline the possible role of CD137L/CD137 signaling in OSCC pathophysiology as well as the potential therapeutic approach.

CD155 is an immune-suppressive ligand that binds to the CD96 and the TIGIT receptors of T-cells. Besides APCs, the CD155 ligand can be expressed on tumor cells, where it can transduce signals to mediate tumor growth and invasion. CD155 overexpression is reported in several malignancies, including NSCSC, esophageal cancer and others [20]. In contrast to these data, the current analysis revealed no significant difference in CD155 expression between OSCC and healthy mucosa. While there was also no significant difference in the expression of TIGIT, there was significant upregulation of CD96 in OSCC tissue. Additionally, the expression levels of the ligand CD155 and its receptor CD96 were strongly correlated in OSCC and weakly correlated in NOM. On the other hand, this correlation was not seen between CD155 and TIGIT. These observations suggest that the CD155–CD96 pathway might be more relevant for OSCC immune evasion compared to TIGIT. Of course, this needs to be further investigated by in vivo and in vitro studies. 

There is evidence for inhibitory functions of CD96, but the exact effect on human T-cells and NK cells is still not fully understood [19]. In human cancer, a correlation of CD96 with T-cell markers and PD-1 was shown. There are contradictory study results regarding the good or poor prognostic influence of high CD96 expression in different human cancer types [19]. These data show that a lot more research is needed to potentially use the CD155-TIGIT/CD96 signaling pathway for cancer immunotherapy. However, the current results show its potential involvement in the immune-suppressive OSCC microenvironment.

GITRL is an activating immune checkpoint ligand that binds to the GITR receptor. These immune modulators are key players in autoimmune diseases and in the immune responses of macrophages [51,52]. In our study, GITRL was significantly upregulated in OSCC tissue compared to NOM. There are no data available regarding GITRL expression in oral cancer. In breast cancer, platelets were identified as a source of GITRL expression besides APCs [53]. Platelet-derived GITRL was higher in breast cancer patients and upregulated during tumor progression [53]. Further research is necessary to understand the role of GITRL signaling in OSCC.

BTLA is an inhibitory receptor of T-cells but is also expressed on B-cells, macrophages and DCs [54]. BTLA binds to herpes virus entry mediator ligand (HVEM) [54], which was not included in the current expression analysis. As BTLA expression is associated with T-cell exhaustion [21], the increased BTLA expression in OSCC tissue might indicate a state of T-cell exhaustion in OSCC. 

No significant difference in expression was detected regarding the immune checkpoint receptors LAG-3, Tim3 and TIGIT. This finding is interesting, as the three immune checkpoints are promising candidates for cancer immunotherapy, with therapeutic substances in the pipeline [14,17,20]. An immunohistochemical analysis in OSCC showed a correlation between high LAG-3 expression and poor survival [55]. In addition, anti-EGFR therapy significantly increased the expression of LAG-3, Tim3, PD-L1 and CTLA-4 [55]. There are several further studies analyzing LAG-3 and Tim3 expression in OSCC [56]. However, especially in relation to Tim3, there are few results available [56].

### 4.3. Limitations of the Study

The NanoString technology has a couple of advantages that were important for the present study. In our study, formalin-fixed, paraffin-embedded material was applied to ensure that the samples were indeed malignant tissue or that the healthy control was indeed NOM. Furthermore, the tissue was dissected to increase the authenticity of the samples. However, the quality of the isolated material was low. The method used does not require high-quality RNA. In addition, only a small amount of starting material is required. It is therefore ideal for scenarios where only small amounts of poor-quality RNA are available.

Besides the advantages, the method has its limitations. This study served to provide an overview of a number of possible immunological changes in tumor tissue compared with NOM, and a number of potential tumor immunologically important modulators of the immune response were identified. Nevertheless, verification of these results with further methods at the mRNA and protein levels is urgently needed. However, expression analysis of all identified candidates using immunohistological and molecular biological assays is very time-consuming and beyond the scope of this study. Verification will thus take place in further studies with external cohorts. In addition, only 20 genes were studied, which were selected based on the current literature. Therefore, this method is not suitable for the discovery of completely new biomarkers and complex signaling pathways. Moreover, the probes used often capture all splice variants of the genes. However, it is known that the genes of many immune modulators produce a variety of isoforms through differential splicing. These transcript variants encode different proteins that may well have different functions in the cell or in the microenvironment. For example, the long isoform of CD8 is membrane-bound, whereas a short isoform is soluble, indicating a completely different function of the protein. Consequently, signaling pathways cannot be determined with certainty by pure expression analysis. Therefore, candidates would not only need to be verified by RT-qPCR and immunohistochemistry, but it would also be necessary to investigate in more detail which transcript variants are expressed and which proteins are actually formed. In addition, in vitro experiments in cell cultures are needed to clarify the function and signaling pathways of the modulators and their individual isoforms. Another pitfall is that the results showing a direct correlation between the expression of receptors and their ligands and the expression of modulators competitively binding to the same protein structures in the same tissue were not validated immunohistochemically by double staining. This needs to be followed up in further studies. 

## 5. Conclusions

The field of tumor immunotherapy is rapidly evolving. In OSCC, the currently observed shift in checkpoint inhibitor use from the palliative to neoadjuvant setting might improve survival. However, the presently available anti-PD-1 immunotherapy will be insufficient for a relevant proportion of patients, even with a multimodal treatment approach involving immunotherapy, surgery and radiotherapy. Therefore, further immune checkpoints besides the PD-L1/PD-1 pathway need to be investigated to make them usable for therapeutic use. However, little data are available regarding immune checkpoint expression in OSCC. For this purpose, we performed a comparative analysis of a large set of checkpoints and immune modulators in OSCC samples and healthy control tissue. We were able to identify several cellular markers, checkpoints and their correlation, as well as their association with histomorphological parameters. As the next steps, verification of these results with further methods at the mRNA and protein levels is necessary. For most of the analyzed immune modulators, functional analyses are also necessary for a better understanding of their role in oral mucosa physiology, malignant transformation and oral cancer progression.

## Figures and Tables

**Figure 1 cancers-14-01812-f001:**
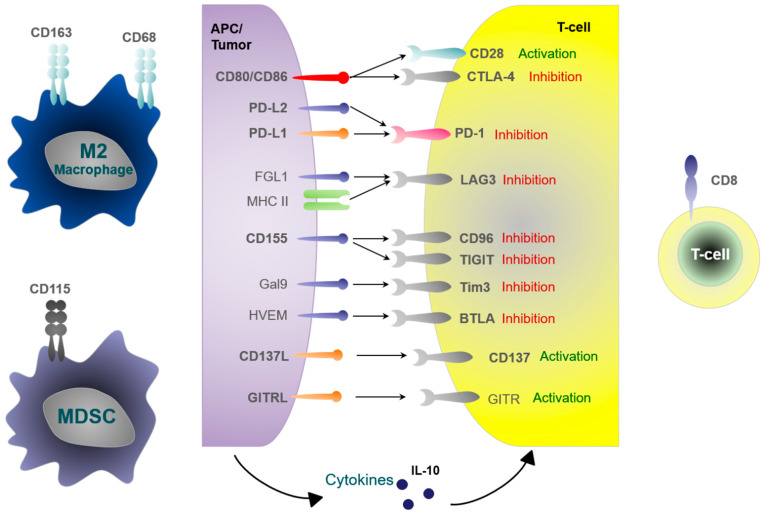
Overview of the cell populations, checkpoint molecules and cytokines analyzed in the study and their interactions to regulate the immune response.

**Figure 2 cancers-14-01812-f002:**
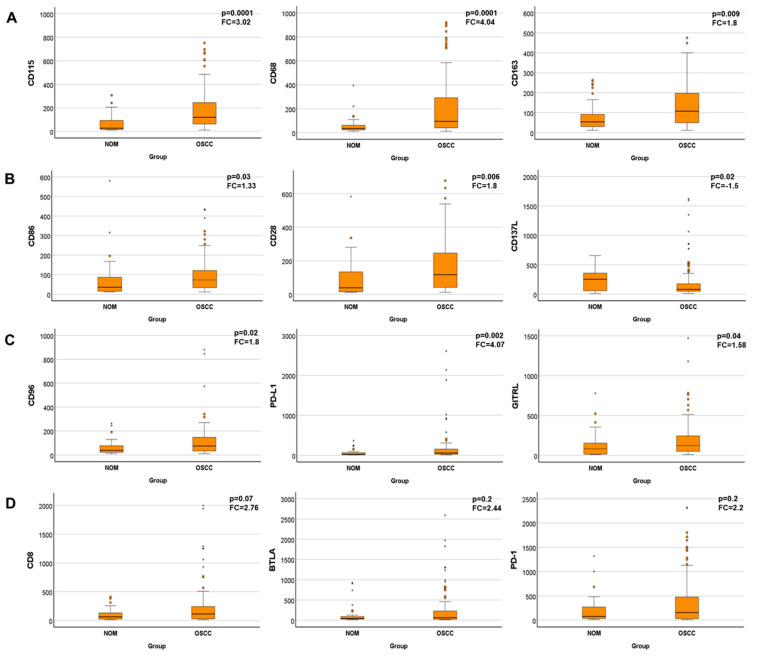
Boxplots of the comparison of expression rates of immune modulators between OSCC and NOM groups. Only results for genes that are significantly deregulated are shown. Analysis of (**A**) specific cell markers for MDSCs (CD115) and macrophages (CD68, CD163), (**B**) activating ligands (CD86, CD28, CD137L), (**C**) inhibitory receptors and ligands (CD96, PD-L1, GIRTL), (**D**) not significantly but highly dysregulated modulators (CD8, BTLA, PD-1). Number of cases of OSCC *n* = 98 and NOM cases *n* = 41. * = extrem value.

**Figure 3 cancers-14-01812-f003:**
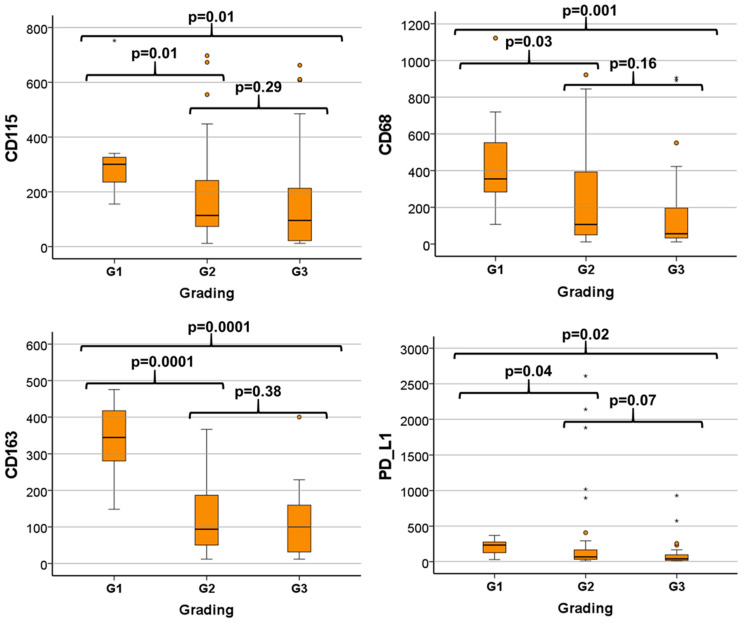
Comparison of expression levels of CD115, CD68, CD163 and PD-L1 between groups of different degrees of differentiation (G1, G2, G3). With increasing dedifferentiation (higher grading), the expression of immune modulators decreased, with the exception of PD-L1. This difference was significant between differentiated (G1) versus undifferentiated malignant tissues (G2 and G3). Number of cases: G1 = 7, G2 = 54, G3 = 35. ○ = outlier; * = extrem value.

**Figure 4 cancers-14-01812-f004:**
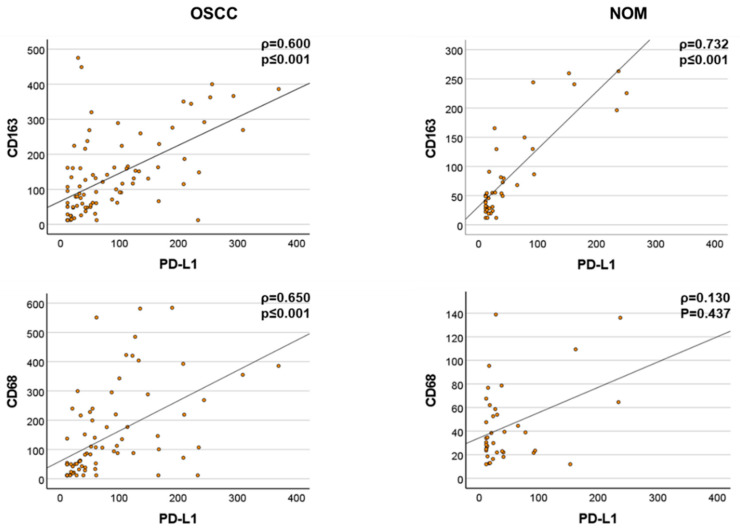
Positive significant correlations between PD-L1 and CD163 expression in the OSCC and NOM groups. Expression levels of CD68 and PD-L1 were strongly correlated in the OSCC (*n* = 98) but not in the NOM group (*n* = 41).

**Figure 5 cancers-14-01812-f005:**
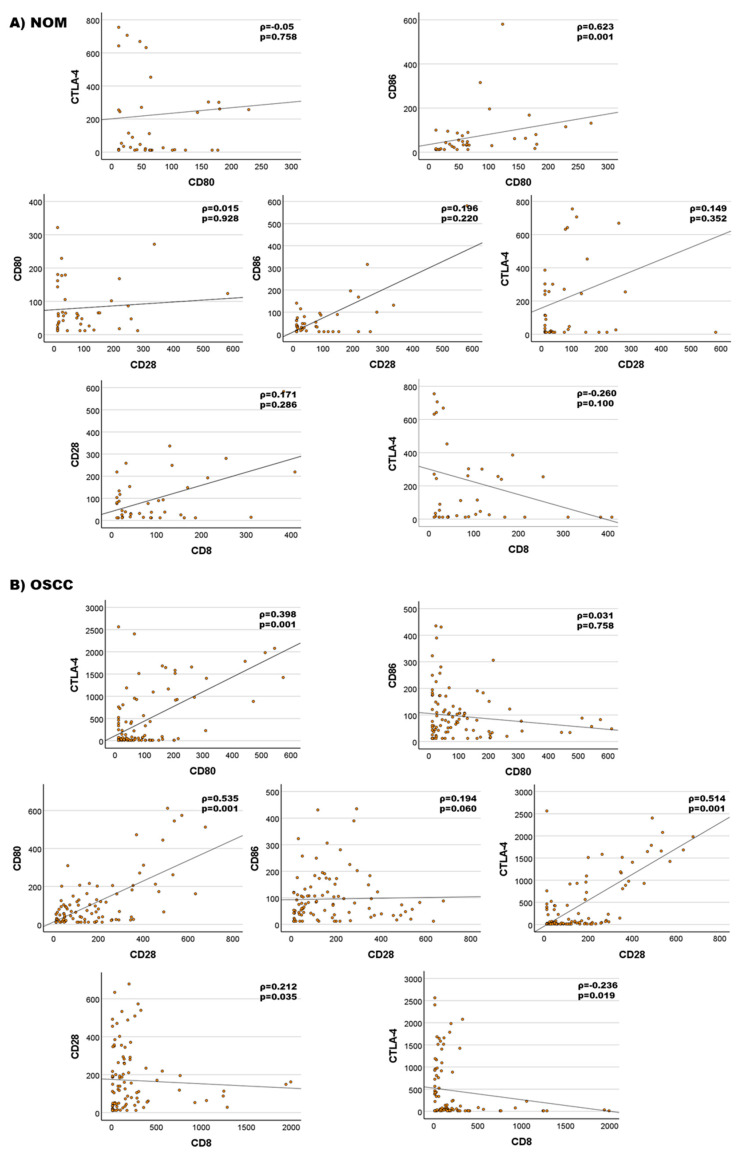
Correlation analysis of the CD28/CTLA-4–CD80/86 signaling pathway and of the immune modulators CD28 and CTLA-4 with the cytotoxic T-cell marker CD8. (**A**) In the NOM group (*n* = 41), only CD80 and CD86 expression levels were moderately correlated. (**B**) OSCC (*n* = 98): strong correlation between CD28 and CD80, e.g., CTLA-4, could be observed. CD80 and CTLA-4 expression levels were moderately correlated. No correlation between CD86 and CD80 was observed.

**Figure 6 cancers-14-01812-f006:**
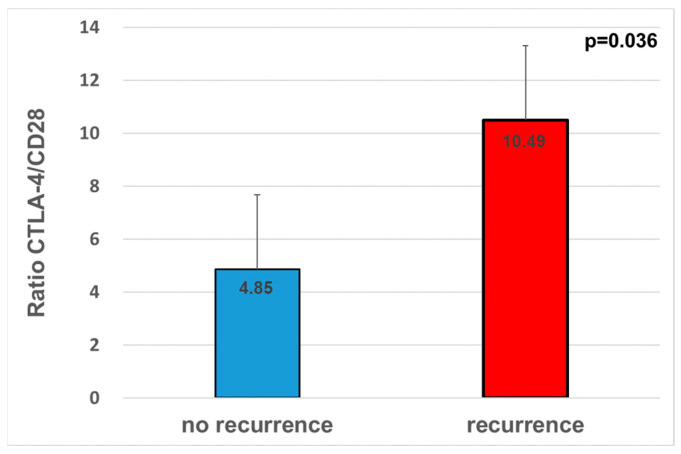
The ratio between CTLA-4 and CD28 was significantly higher in OSCC patients suffering from lymph node metastasis. Cases of no recurrence *n* = 62. Number of patients suffering from recurrence *n* = 25.

**Table 1 cancers-14-01812-t001:** nCounter PlexSet Design and Isoform Coverage. * Name/alternative gene symbol; ** HUGO gene by HGNC = HUGO Gene Nomenclature Committee; *** used for probe design; § number of isoforms detected by the probe.

Immune Modulation	Gene Symbol *	HUGO Gene **	Accession ***	Position	Hits §
Myeloid Cell	CD115	CSF1R	NM_005211.2	3776–3875	4
Macrophage	CD68	CD68	NM_001251.2	1141–1240	2
Macrophage	CD163	CD163	NM_004244.4	1631–1730	5
Cytotoxic T-cell	CD8	CD8A	NM_001768.5	1321–1420	4
Activating Receptor	CD80	CD80	NM_005191.3	675–774	3
Activating Receptor	CD86	CD86	NM_175862.3	1266–1365	5
Activating Receptor	CD137	TNFRSF9	NM_001561.4	256–355	3
Activating Ligand	CD137L	TNFSF9	NM_003811.3	399–498	1
Activating Ligand	CD28	CD28	NM_001243078.1	2066–2165	6
Inhibitory Receptor	CTLA-4	CTLA-4	NM_005214.3	406–505	2
Inhibitory Receptor	LAG-3	LAG-3	NM_002286.5	1736–1835	2
Inhibitory Receptor	TIM3	HAVCR2	NM_032782.3	956–1055	1
Inhibitory Receptor	TIGIT	TIGIT	NM_173799.2	1969–2068	3
Inhibitory Receptor	CD96	CD96	NM_005816.4	429–528	12
Inhibitory Receptor	BTLA	BTLA	NM_181780.2	306–405	4
Inhibitory Receptor	PD-1	PDCD1	NM_005018.2	311–410	3
Inhibitory Ligand	PD-L1	CD274	NM_014143.3	50–149	4
Inhibitory Ligand	PD-L2	PDCD1LG2	NM_025239.3	236–335	2
Inhibitory Ligand	CD155	PVR	NM_006505.3	605–704	4
Inhibitory Ligand	GITRL	TNFSF18	NM_005092.2	176–275	1
Inhibitory Cytokine	IL-10	IL10	NM_000572.2	231–330	1
Endogene Control	Polymerase II	POLR2A	NM_000937.2	3776–3875	1
Endogene Control	RPL19	RPL19	NM_000981.3	316–415	2
Endogene Control	GAPDH	GAPDH	NM_001256799.1	387–486	6

**Table 2 cancers-14-01812-t002:** Description of the study collective. Total number of cases: 139. * Grouped. The two groups of the collective were gender-matched (*p* = 0.27) but not age-matched (*p* = 0.01). DFS = disease-free survival; OS = overall survival.

Clinical and Histopathological Parameters		Patients		Healthy Volunteers	
		*n*	% of Cases	*n*	% of Cases
Number of cases		98		41	
Gender	Male	71	72.4	26	63.4
	Female	27	27.6	15	36.6
Mean age ± SD		63.1 ± 11.7 years		49.2 ± 19.3 years	
Range of age		35–93 years		18–67 years	
Tumor status	T1/T2	58	59.2		
	T3/T4	39	39.8		
	Unknown	1	1.0		
N-Status *	N0	51	52.0		
	N1	47	48.0		
Grading	G1	7	7.1		
	G2	54	55.1		
	G3	35	35.7		
	Unknown	2	2		
Clinical stage *	Early	33	33.7		
	Late	64	65.3		
	Unknown	1	1		
Recurrence	No	62	63.3		
	Yes	25	25.5		
	Unknown	11	11.2		
Life status 9/2021	Alive	54	55.1		
	Dead	24	24.5		
	Unknown	20	20.4		
DFS (months)	Mean ± SD	12.68 ± 16.48			
	Range	1–71			
OS (months)	Mean ± SD	31.5 ± 27.1			
	Range	1–112			

**Table 3 cancers-14-01812-t003:** Statistical analysis of differential expression of immune modulators in OSCC (*n* = 98) compared to NOM group (*n* = 41). Mean values of expression, expression differences (FC) and statistical significance expressed as *p*-values (MWU test) are reported. * Average of counts in the group. Statistically relevant changes are highlighted in bold. The genes are over-/under-expressed about two fold, but the change of differential eypresion do not reach statistic relavance between the compared groups.

Immune Modulators
Group	CD115	CD68	CD163	CD8	CD80	CD86	CD137
	Myeloid cell	Macrophage	CytotoxicT-cell	Activating receptor
NOM *	66.33	57.06	83.29	95.29	80.48	71.83	241.37
OSCC *	200.13	230.37	148.88	263.39	104.1	95.43	261.84
FC	**3.02**	**4.04**	**1.79**	**2.76**	1.29	**1.33**	−1.1
*p*-value	**0.0001**	**0.0001**	**0.009**	0.07	0.95	**0.03**	0.22
Up/down-regulation	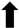	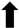	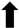	⇑		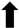	
Group	CD137L	CD28	GITRL	CTLA-4	LAG-3	Tim3	TIGIT
	Activating ligand	Inhibitory receptor
NOM *	249.73	96.79	129.50	228.42	88.74	149.31	392.41
OSCC *	208.53	169.19	217.98	455.27	85.21	196.98	423.04
FC	**−1.50**	**1.75**	**1.58**	**1.99**	−1.04	1.32	1.11
*p*-value	**0.017**	**0.006**	**0.04**	0.32	0.49	0.4	0.15
Up/down-regulation	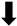	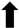	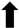	⇑			
Group	CD96	BTLA	PD−1	PD-L1	PD-L2	CD155	IL-10
	Inhibitory receptor	Inhibitory ligand	Inhibitorycytokine
NOM *	62.79	122.09	195.38	60.21	177.56	162.55	116.25
OSCC *	112.16	297.89	361.59	191.09	221.11	131.37	136.35
FC	**1.79**	**2.44**	**2.20**	**4.07**	1.35	−1.24	1.17
*p*-value	**0.02**	0.20	0.20	**0.002**	0.26	0.22	0.46
Up/down-regulation	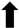	⇑	⇑	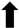			

**Table 4 cancers-14-01812-t004:** Average of counts in specific groups, changes in expression levels (FC) and statistical relevance of differential expression in OSCC tissues depending on tumor size, lymph node status (N), clinical stage (UICC), grading (G) and recurrence (Rec.). Number of OSCC = 98. Statistically relevant changes are highlighted in bold. All significances are calculated by means of the MWU test with the exception of grading (* Kruskal–Wallis Test). FC significance level was 1.5 times. Statistically relevant changes are highlighted in bold.

		CD115	CD68	CD163	CD8	CD80	CD86	CD137
		Myeloid cell	Macrophage	CytotoxicT-cell	Activating receptor
T-Status	**T1/T2**	237.44	261.30	152.79	309.80	117.63	100.39	233.08
**T3/T4**	149.47	192.66	147.03	205.99	84.46	92.21	299.31
**FC**	**−1.6**	−1.3	−1.1	−1.5	−1.4	−1.1	1.3
***p*-value**	0.05	0.04	0.09	0.26	0.47	0.22	0.49
	**Regulation** **T3/T4**	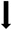	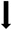					
N-Status	**N0**	203.71	178.85	139.70	307.36	93.16	90.96	182.87
**N1**	199.53	290.12	161.76	221.10	117.84	102.05	345.55
**FC**	1.0	**1.6**	1.2	−1.4	1.3	1.1	**1.9**
***p*-value**	0.32	0.05	0.58	0.46	0.62	0.2	0.24
	**Regulation** **N1**		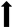					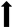
Grading	**G1**	333.15	465.30	337.72	270.75	197.23	89.47	265.80
**G2**	211.90	256.98	146.15	223.18	103.27	102.06	211.80
**G3**	165.39	157.54	123.50	344.06	86.68	92.46	329.32
**FC**	n.d.	n.d.	n.d	n.d.	n.d.	n.d.	n.d.
***p*-value ***	**0.02**	**0.01**	**0.001**	0.48	0.36	0.65	0.81
UICC	**early**	233.68	251.03	157.92	355.74	83.05	91.93	143.99
**late**	185.78	243.33	146.64	222.86	115.25	99.77	319.38
**FC**	−1.3	1.1	−1.1	**−1.6**	1.4	1.1	**2.2**
***p*-value**	0.74	0.59	0.49	0.41	0.31	0.86	0.16
	**Regulation** **late**				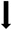			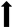
Rec.	**No**	209.22	233.05	145.78	283.90	**107.02**	96.66	258.79
**Yes**	199.17	251.83	171.09	291.94	48.67	111.58	171.93
**FC**	1	1		1	**−2.2**	1.2	**−1.5**
***p*-value**	0.68	0.28	0.74	0.65	**0.02**	0.24	0.71
	**Regulation** **yes**					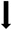		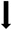
		**CD137L**	**CD28**	**GITRL**	**CTLA-4**	**LAG-3**	**Tim3**	**TIGIT**
		Activating ligand	Inhibitory receptor
T-Status	**T1/T2**	252.02	183.14	217.49	483.97	90.02	242.43	447.09
**T3/T4**	152.38	144.52	224.59	393.42	81.53	138.01	367.85
**FC**	**−1.7**	−1.3	1.0	−1.2	−1.1	**−1.7**	−1.2
***p*-value**	**0.036**	0.197	0.34	0.07	0.93	0.18	0.21
	**Regulation** **T3/T4**	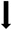					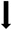	
N-Status	**N0**	169.27	159.33	211.03	403.67	63.77	208.22	453.33
**N1**	255.31	183.03	229.90	520.70	109.80	188.72	425.63
**FC**	**1.5**	1.1	1.1	1.3	**1.7**	−1.1	−1.1
***p*-value**	0.26	0.52	0.91	0.3	0.14	0.7	0.53
	**Regulation** **N1**							
Grading	**G1**	186.57	224.84	289.91	630.41	68.98	95.73	529.87
**G2**	260.14	177.37	211.43	476.63	90.36	226.32	367.39
**G3**	147.78	144.92	225.25	378.58	86.47	186.62	476.45
**FC**	n.d.	n.d.	n.d.	n.d.	n.d.	n.d.	n.d.
***p*-value ***	0.34	0.58	0.47	0.08	0.89	0.37	0.78
UICC	**early**	169.55	147.15	241.62	357.58	68.88	240.77	477.93
**late**	233.83	178.16	209.38	493.96	95.75	179.71	382.90
**FC**	1.4	1.2	−1.2	1.4	1.4	−1.3	−1.2
***p*-value**	0.69	0.37	0.71	0.49	0.73	0.48	0.35
	**Regulation** **late**							
Rec	**No**	212.64	169.89	189.34	453.72	93.61	206.71	437.73
**Yes**	267.05	124.61	285.43	214.51	97.03	255.92	380.52
**FC**	1.3	−1.4	**1.5**	**−2.1**	1.0	1.2	−1.2
***p*-value**	0.09	0.52	0.79	0.07	0.54	0.39	0.09
	**Regulation** **yes**			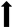	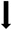			
		**CD96**	**BTLA**	**PD-1**	**PD-L1**	**PD-L2**	**CD155**	**IL-10**
		Inhibitory receptor	Inhibitory ligand	Inhibitory cytokine
T-Status	**T1/T2**	135.37	278.95	425.59	372.51	232.69	129.53	154.41
**T3/T4**	81.95	338.88	452.99	65.19	250.10	137.64	114.85
**FC**	**−1.7**	1.2	1	**−5.7**	1.1	1.1	−1.3
***p*-value**	0.08	0.27	0.72	**0.001**	0.85	0.24	**0.009**
	**Regulation** **T3/T4**	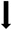			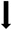			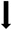
N-Status	**N0**	91.66	294.38	293.89	320.80	265.62	111.91	134.12
**N1**	136.34	307.79	583.78	168.32	215.25	153.43	141.42
**FC**	**1.5**	1.0	**2.0**	**−1.9**	−1.2	1.4	1.1
***p*-value**	0.79	3.8	0.39	0.78	0.23	0.82	0.77
	**Regulation** **N1**				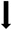			
Grading	**G1**	145.79	77.25	184.38	206.35	204.03	248.97	136.41
**G2**	118.11	226.38	426.72	354.03	212.91	130.09	116.86
**G3**	103.91	474.81	501.37	101.79	294.53	114.56	175.93
**FC**	n.d.	n.d.	n.d.	n.d.	n.d.	n.d.	n.d.
***p*-value**	0.13	0.51	0.72	**0.03**	0.28	0.17	0.91
UICC	**Early**	91.99	312.56	302.53	452.13	242.13	126.50	152.51
**Late**	125.18	298.14	505.74	144.18	238.44	136.03	131.29
**FC**	1.4	1.0	**1.7**	−3.1	1.0	1.1	−1.2
***p*-value**	0.73	0.33	0.47	0.33	0.63	0.14	0.3
	**Regulation** **late**							
Rec	**No**	117.26	340.27	426.41	321.03	241.84	120.15	127.63
**Yes**	96.95	227.35	472.44	129.76	196.25	186.73	137.79
**FC**	−1.2	−1.5	1.1	−2.5	−1.2	**1.6**	1.1
***p*-value**	0.67	0.28	0.86	0.48	0.33	0.35	0.17
	**Regulation** **yes**						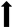	

**Table 5 cancers-14-01812-t005:** Closer analysis of the changes in the expression levels of immune checkpoints, which showed significant differences in the * Kruskal–Wallis (KW) test, in relation to the increase in dedifferentiation (grading). Number of cases: G1 = 7, G2 = 54, G3 = 35. Statistically relevant changes are highlighted in bold.

A: Tissue		CD115	CD68	CD163	PD-L1
		Myeloid cell	Macrophage		Inhibitory ligand
**Grading ***		**0.02**	**0.01**	**0.001**	**0.03**
**G1 vs. G2**	G1	333.15	465.30	337.72	206.35
	G2	211.90	256.98	146.15	354.03
	FC	1.6	1.8	2.3	−1.7
	*p*-value	0.01	0.03	<0.0001	0.04
	Regulation	up	up	up	down
**G1 vs. G3**	G1	333.15	465.30	337.72	206.35
	G3	165.39	157.54	123.50	101.79
	FC	2.0	2.95	2.7	2.0
	*p*-value	0.01	0.001	<0.0001	0.02
	Regulation	up	up	up	up
**G2 vs. G3**	G2	211.90	256.98	146.15	354.03
	G3	165.39	157.54	123.50	101.79
	FC	1.28	1.6	1.2	3.5
	*p*-value	0.29	0.16	0.38	0.07
	Regulation	-	-	-	up

**Table 6 cancers-14-01812-t006:** Correlation of CD163 and CD68 expression with PD ligands. (**A**) All samples were used. (**B**) OSCC group. (**C**) NOM group. The *p*-value was calculated by Spearman’s rho correlation. ρ = Correlation coefficient; *n* = number of observations correlated. Statistically relevant changes are highlighted in bold.

Immune Modulator	Correlation	CD163	CD68	PD-L1	PD-L2
**(A) All**					
CD163	ρ	1.000			
	*p*-value				
	*n*	139			
CD68	ρ	**0.704**	1.000		
	*p*-value	**<0.001**			
	*n*	139	139		
PD-L1	ρ	**0.606**	**0.560**	1.000	
	*p*-value	**<0.001**	**<0.001**		
	*n*	139	139	139	
PD-L2	ρ	**0.216**	−0.060	**0.257**	1.000
	*p*-value	**0.010**	0.482	**0.002**	
	*n*	139	139	139	139
**(B) NOM**	Correlation	CD163	CD68	PD-L1	PD-L2
CD163	ρ	1.000			
	*p*-value				
	*n*	41			
CD68	ρ	**0.404**	1.000		
	*p*-value	**0.009**			
	*n*	41	41		
PD-L1	ρ	**0.739**	0.130	1.000	
	*p*-value	**<0.001**	0.437		
	*n*	41	41	41	
PD-L2	ρ	0.278	0.021	**0.469**	1.000
	*p*-value	0.079	0.894	**0.002**	
	*n*	41	41	41	41
**(C) OSCC**	Correlation	CD163	CD68	PD-L1	PD-L2
CD163	ρ	1.000			
	*p*-value				
	*n*	98			
CD68	ρ	**0.737**	1.000		
	*p*-value	**<0.001**			
	*n*	98	98		
PD-L1	ρ	**0.600**	**0.650**	1.000	
	*p*-value	**<0.001**	**<0.001**		
	*n*	98	98	98	
PD-L2	ρ	0.113	−0.158	0.176	1.000
	*p*-value	0.264	0.119	0.081	
	*n*	98	98	98	98

**Table 7 cancers-14-01812-t007:** Correlation of CD28, CD80, CD86 and CTLA-4 expression and correlation of CD8 with CD28 and CTLA-4. (**A**) All samples were used (*n* = 139). Only (**B**) tumor (*n* = 98) and (**C**) NOM (*n* = 41) tissues were taken into account. The *p*-value was calculated by Spearman’s rho correlation. * The correlation is significant at the 0.01 level (2-tailed). ρ = Correlation coefficient; *n* = number of observations correlated. Statistically relevant changes are highlighted in bold.

	Correlation	CD28	CD80	CD86	CTLA-4
**A. All**					
CD28	ρ	1.000			
	*p*-value				
CD80	ρ	**0.393 ***	1.000		
	*p*-value	**0.001**			
CD86	ρ	**0.289 ***	**0.196 ***	1.000	
	*p*-value	**0.001**	**0.02**		
CTLA-4	ρ	**0.405 ***	**0.291 ***	−0.08	1.000
	*p*-value	**0.001**	**0.001**	0.352	
CD8	ρ	**0.269 ***	n.d.	n.d.	**−0.227 ***
	*p*-value	**0.001**	n.d.	n.d.	**0.007**
**B. OSCC**					
CD28	ρ	**1.000**			
	*p*-value				
CD80	ρ	**0.535 ***	1.000		
	*p*-value	**0.001**			
CD86	ρ	0.194	0.031	1.000	
	*p*-value	0.06	0.758		
CTLA-4	ρ	**0.514 ***	**0.398 ***	−0.58	1.000
	*p*-value	**0.001**	**0.001**	0.571	
CD8	ρ	**0.212 ***	n.d.	n.d.	**−0.236 ***
	*p*-value	**0.035**	n.d.	n.d.	**0.019**
**C. NOM**					
CD28	ρ	1.000	0.015		
	*p*-value		0.928		
CD80	ρ	0.015	1.000	**0.623 ***	
	*p*-value	0.928		**0.001**	
CD86	ρ	0.196	**0.623 ***	1.000	
	*p*-value	0.220	**0.001**		
CTLA-4	ρ	0.149	−0.05	−0.224	1.000
	*p*-value	0.352	0.758	0.159	
CD8	ρ	0.171	n.d.	n.d.	−0.260
	*p*-value	0.286	n.d.	n.d.	0.100

**Table 8 cancers-14-01812-t008:** Expression of immune-modulatory proteins relative to cellular markers in OSCC and NOM. (**a**) Expression relative to cytotoxic T-cell marker (CD8), (**b**) to the MDSC marker CD115 and (**c**) to macrophage marker CD68. (**d**) CD163 Marker for M2 polarization.

a				
Ratio	NOM	OSCC	Quotient (OSCC/NOM)	*p*-Value
CTLA-4/CD8	10.04	13.98	1.39	0.58
LAG-3/CD8	1.14	0.65	0.57	0.001
PD-1/CD8	2.47	10.89	4.41	0.26
TIGIT/CD8	9.16	4.16	0.45	0.001
CD96/CD8	1.40	2.10	1.50	0.95
TIM3/CD8	2.94	5.54	1.88	0.08
BTLA/CD8	1.48	6.32	4.27	0.68
CD28/CD8	2.32	3.48	1.50	0.44
CD137/CD8	4.43	7.79	1.76	0.01
CD80/CD8	1.35	1.73	1.28	0.141
**b**				
**Ratio**	**NOM**	**OSCC**	**Quotient (OSCC/NOM)**	***p*-Value**
CD163/CD115	1.79	1.63	0.91	0.001
CD68/CD115	1.63	1.68	1.03	0.71
PDL1/CD115	1.06	3.37	3.18	0.025
PDL2/CD115	4.65	2.88	0.62	0.0001
CD80/CD115	2.12	1.19	0.56	0.0001
CD86/CD115	1.74	1.41	0.81	0.004
GITRL/CD115	4.94	5.95	1.20	0.029
CD137L/CD115	10.49	4.70	0.45	0.0001
CD155/CD115	3.75	1.62	0.43	0.0001
**c**				
**Ratio**	**NOM**	**OSCC**	**Quotient (OSCC/NOM)**	***p*-Value**
CD163/CD68	2.22	1.43		0.003
CD80/CD68	2.20	1.40	1.57	0.002
CD86/CD68	1.91	1.97	1.0	0.043
GITRL/CD68	4.04	3.25	0.8	0.248
CD137L/CD68	6.92	2.02	0.29	0.0001
CD155/CD68	4.52	1.40	0.31	0.0001
PDL1/CD68	0.73	4.30	5.89	0.321
PDL2/CD68	5.98	6.15	1.03	0.019
**d**				
**Ratio**	**NOM**	**OSCC**	**Quotient (OSCC/NOM)**	***p*-Value**
CD80/163	1.19	1.09	0.92	0.142
CD86/163	1.07	1.43	1.34	0.704
GITRL/163	3.06	2.61	0.85	0.712
CD137L/CD163	5.27	2.43	0.46	0.0001
CD155/Cd163	2.20	1.43	0.65	0.0001
PDL1/CD163	0.74	4.32	5.84	0.127
PDL2/CD163	3.77	4.26	1.13	0.175

## Data Availability

All the data can be accessed in main text and Appendix A.

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
