# Peer review of "Beyond PD-L1—Identification of Further Potential Therapeutic Targets in Oral Cancer"

_cancers, 2022, doi:10.3390/cancers14071812_

Round 1

Reviewer 1 Report

In present research, authors perform the analysis of immune cells in OSCC and healthy controls and determine its prognostic influence. The landscape of immune cells is used to predict the efficacy of immunotherapy treatment. Although t is an interesting study, I have several reservations, my comments are appended as below:

Major comments:

  1. Information on inhibitory receptors in introduction- this should be a single para, not fragmented one.
  2. While specifying malignancies, authors should specify exact type. For instance, reference 24,25,26. This should be followed in complete manuscript.
  3. Selection of patients: authors should include a flowchart along with inclusion and exclusion criterion.
  4. Figure 1-4- include n in figure.
  5. Most important point is validation of profiling is missing; authors should use either cell lines or external cohorts.
  6. Authors attempt to do IHC for receptors with direct correlation in same tissue. This may validate in part the observations.
  7. Authors should note limitations of this study.

Minor comments:

  1. Introduction- authors should first discuss on the disease statistics, present standard of care.
  2. For immunotherapy, other cofounders as BMI, smoking are known to play an important role. Authors may refer PMID: 33076303and add a para.
  3. catalogue no of all used reagents should be noted.

4. Manuscript should be refined for language correctness

Author Response

Point 1: Information on inhibitory receptors in introduction- this should be a single para, not fragmented one.

Response 1: The information has been summarised in one paragraph.

Point 2: While specifying malignancies, authors should specify exact type. For instance, reference 24,25,26. This should be followed in complete manuscript.

Response 2:

We included the analysed tumor entities as you recommended:

Introduction:

Besides immune checkpoints, the cellular infiltrate in human malignancies including HNSCC is highly relevant for tumor progression, response to therapies and survival [1]. In this regard, T-cells, macrophages and myeloid derived suppressor cells (MDSC) are of special interest. CD8 positive T-cells are the main effector cells of anti-tumor immunity [2]. Various tumor entities (including NSCLC, melanoma and others) with a high CD8 infiltration were shown to have a better response to immunotherapy [2]. However, a recent meta-analysis showed no association with CD8 infiltration and survival in oral cancer [3].The infiltration of tumor-associated CD68 positive macrophages and in particular the tumor-promoting, M2-polarized, CD163 expressing macrophages were shown to be associated with oral cancer initiation and tumor progression [4-6].

Discussion:

In animal models, CD137 activating antibodies showed anti-tumor activity [7]. Currently, there are several early human trials ongoing especially in combination with other checkpoint inhibitors in several malignancies including breast cancer, colon cancer and other advanced malignancies [7-9].

CD155 is an immune suppressive ligand that binds to the CD96 and TIGIT receptor of T-cells. Besides APCs, the CD155 ligand can be expressed on tumor cells where it can signal to mediate tumor growth and invasion. CD155 overexpression is reported in several malignancies including NSCSC, esophageal cancer and others [10].

Point 3: Selection of patients: authors should include a flowchart along with inclusion and exclusion criterion.

Response 3:  The criteria for inclusion of patients and healthy volunteers into the study are already described in the material and methods section (page 4): “Patients were only included if it was an initial diagnosis of oral malignancy and no previous treatment was given. The selection of healthy subjects was based on the absence of tumor disease, general disease, and acute or chronic inflammatory conditions of the normal oral mucosa.”

Point 4: Figure 1-4- include n in figure.                                        

Response 4: N is Included in legends of all figures now.

Point 5: Most important point is validation of profiling is missing; authors should use either cell lines or external cohorts.

Response 5:

Today there is only little data available regarding immune checkpoint expression in OSCC. This study was aimed to perform a comparative analysis of a large number of immune checkpoints in OSCC compared to normal oral mucosa in order to provide a first large overview of possible immunological changes in oral tumors (Immunoscore) and to identify several potential targets for therapeutic application. Thus, the differential expression of 20 immune checkpoints, both immunosuppressive and co-stimulatory, were analysed by a Nanostring Assay in advance. Possible tumour-immunologically important modulators of the immune response were identified. As next steps, a verification of these results with further methods on mRNA- and protein-level is necessary. In addition, the mechanism of immunomodulation controlled by immune checkpoint interactions has to be further elucidated for a better understanding of their role in oral mucosa physiology, malignant transformation and oral cancer progression. However, the verification of all identified candidates via immunohistological and molecular biological tests is very time-consuming. The verification will now be done in further investigations using external cohorts.

This problem is now addressed in the section “limitations of the study”:

“This study served to provide an overview of a number of possible immunological changes in tumor tissue compared with NOM, and a number of potential tumor immunologically important modulators of the immune response were identified. However, verification of these results with further methods at the mRNA and protein levels is urgently needed. However, expression analysis of all identified candidates by immunohistological and molecular biological assays is very time-consuming and beyond the scope of this study. Verification will now take place in further studies with external cohorts.”

Point 6: Authors attempt to do IHC for receptors with direct correlation in same tissue. This may validate in part the observations.

Response 6: This study provided the first insights of possible correlations between the altered expression levels of receptors and their ligands, as well as the expression of modulators that bind competitively to the same protein structures. For instance, a direct correlation between the expression of the immune modulators, CD80/86 and CTLA-4 and CD28, could be observed. In further studies, the expression of these receptors and ligands in the same tissue will be investigated immunologically by means of double staining in order to validate the correlation of their altered expression levels in tumors. This problem is now addressed in the section “limitations of the study”.

“Another pitfall is that the results showing a direct correlation of the expression of receptors and their ligands and the expression of modulators competitively binding to the same protein structures in the same tissue have not been validated immunohistochemically by double staining. This needs to be made up in further studies.”

Point 7: Authors should note limitations of this study.

Response 7: A Chapter is inserted into discussion.

Limitations of the Study

The NanoString technology has a couple of advantages that were important for the present study.  In our study, formalin-fixed, paraffin-embedded material was applied to ensure that the samples were indeed malignant tissue or that the healthy control was indeed NOM. Furthermore, the tissue was dissected to increase the authenticity of the samples. However, the quality of the isolated material is low. The method used does not require high quality RNA. In addition, only a small amount of starting material is required. It is therefore ideal for scenarios where only small amounts and poor quality RNA are available.

Besides the advantages, the method has its limitations. This study served to provide an overview of a number of possible immunological changes in tumor tissue compared with NOM, and potential tumor immunologically important modulators of the immune response were identified. Nevertheless, verification of these results with further methods at the mRNA and protein levels is urgently needed. However, expression analysis of all identified candidates by immunohistological and molecular biological assays is very time-consuming and beyond the scope of this study. Verification will now take place in further studies with external cohorts. Additionally, only 20 genes were studied, which were selected based on the current literature. Therefore, this method is not suitable for the discovery of completely new biomarkers and complex signalling pathways. Moreover, the probes used often capture all splice variants of the genes. However, it is known that the genes of many immunomodulators produce a variety of isoforms through differential splicing. These transcript variants encode different proteins that may well have different functions in the cell or in the microenvironment. For example, the long isoform of CD8 is membrane-bound, whereas a short isoform is soluble, indicating a completely different function of the protein. Consequently, signalling pathways cannot be determined with certainty by pure expression analysis. Therefore, candidates would not only need to be verified by RT-qPCR and immunohistochemistry, but it would also be necessary to investigate in more detail which transcript variants are expressed and which proteins are actually formed. In addition, in vitro experiments in cell cultures are needed to clarify the function and signalling pathways of the modulators and their individual isoforms. Another pitfall is that the results showing a direct correlation of the expression of receptors and their ligands and the expression of modulators competitively binding to the same protein structures in the same tissue have not been validated immunohistochemically by double staining. This needs to be made up in further studies.

Minor comments:

Point 8:  Introduction- authors should first discuss on the disease statistics, present standard of care.

Response 8: Oral squamous cell carcinoma (OSCC) is the most common tumor of the oral cavity, accounting for more than 90% [11]. In 2020, there were 377,713 cases of lip and oral cavity carcinoma worldwide, making the disease the 5th most common carcinoma. 177,757 of the patients with the disease died [12].

Point 9: For immunotherapy, other cofounders as BMI, smoking are known to play an important role. Authors may refer PMID: 33076303and add a para.[13]

Response 9: Besides the immune microenvironment, many other players are responsible for the efficacy of immunotherapies. A review article by Deshpande et al lists these and describes their influence. In our study, however, we want to directly identify modulators that influence the immune system in order to test their utility for the development of new immunotherapies. Therefore, the authors would like to omit the description of other cofounders, which play an important role in the application of e.g. immune checkpoint inhibitors, but are not subjects of this study.

Point 10: catalogue no of all used reagents should be noted.

Response 10: The catalogue number of all reagents used have been inserted (Qiagen, Hilden, Germany, cat.no. 217504). The panel of the nCounter™ assay was made on demand for customers` solutions. Therefore no catalogue No is available.

Point 11:  Manuscript should be refined for language correctness

Response 11: The manuscript was carefully read and corrected by the authors again.

Additional literature

  1. Mazilu L, Suceveanu AI, Stanculeanu DL, Gheorghe AD, Fricatel G, Negru SM: Tumor microenvironment is not an 'innocent bystander' in the resistance to treatment of head and neck cancers (Review). Exp Ther Med 2021, 22(4):1128.
  2. Li F, Li C, Cai X, Xie Z, Zhou L, Cheng B, Zhong R, Xiong S, Li J, Chen Z et al: The association between CD8+ tumor-infiltrating lymphocytes and the clinical outcome of cancer immunotherapy: A systematic review and meta-analysis. EClinicalMedicine 2021, 41:101134.
  3. Borsetto D, Tomasoni M, Payne K, Polesel J, Deganello A, Bossi P, Tysome JR, Masterson L, Tirelli G, Tofanelli M et al: Prognostic Significance of CD4+ and CD8+ Tumor-Infiltrating Lymphocytes in Head and Neck Squamous Cell Carcinoma: A Meta-Analysis. Cancers (Basel) 2021, 13(4).
  4. Bruna F, Scodeller P: Pro-Tumorigenic Macrophage Infiltration in Oral Squamous Cell Carcinoma and Possible Macrophage-Aimed Therapeutic Interventions. Front Oncol 2021, 11:675664.
  5. Weber M, Wehrhan F, Baran C, Agaimy A, Buttner-Herold M, Ozturk H, Neubauer K, Wickenhauser C, Kesting M, Ries J: Malignant transformation of oral leukoplakia is associated with macrophage polarization. J Transl Med 2020, 18(1):11.
  6. Weber M, Buttner-Herold M, Hyckel P, Moebius P, Distel L, Ries J, Amann K, Neukam FW, Wehrhan F: Small oral squamous cell carcinomas with nodal lymphogenic metastasis show increased infiltration of M2 polarized macrophages--an immunohistochemical analysis. J Craniomaxillofac Surg 2014, 42(7):1087-1094.
  7. Etxeberria I, Glez-Vaz J, Teijeira A, Melero I: New emerging targets in cancer immunotherapy: CD137/4-1BB costimulatory axis. ESMO Open 2020, 4(Suppl 3):e000733.
  8. Ye L, Jia K, Wang L, Li W, Chen B, Liu Y, Wang H, Zhao S, He Y, Zhou C: CD137, an attractive candidate for the immunotherapy of lung cancer. Cancer Sci 2020, 111(5):1461-1467.
  9. Hashimoto K: CD137 as an Attractive T Cell Co-Stimulatory Target in the TNFRSF for Immuno-Oncology Drug Development. Cancers (Basel) 2021, 13(10).
  10. Ge Z, Peppelenbosch MP, Sprengers D, Kwekkeboom J: TIGIT, the Next Step Towards Successful Combination Immune Checkpoint Therapy in Cancer. Front Immunol 2021, 12:699895.
  11. Miguelanez-Medran BC, Pozo-Kreilinger JJ, Cebrian-Carretero JL, Martinez-Garcia MA, Lopez-Sanchez AF: Oral squamous cell carcinoma of tongue: Histological risk assessment. A pilot study. Med Oral Patol Oral Cir Bucal 2019, 24(5):e603-e609.
  12. Sung H, Ferlay J, Siegel RL, Laversanne M, Soerjomataram I, Jemal A, Bray F: Global Cancer Statistics 2020: GLOBOCAN Estimates of Incidence and Mortality Worldwide for 36 Cancers in 185 Countries. CA Cancer J Clin 2021, 71(3):209-249.
  13. Deshpande RP, Sharma S, Watabe K: The Confounders of Cancer Immunotherapy: Roles of Lifestyle, Metabolic Disorders and Sociological Factors. Cancers (Basel) 2020, 12(10).

Reviewer 2 Report

Green typos in the text.

Some opinions to improve the overall quality:

Introduction

  • the current surgical approach however also includes the use of innovative methods such as robotic surgery which can provide not only an approach in R0 but also in salvage surgery allowing to de-intensify adjuvant chemoradiotherapy, please cite doi:10.1016/j.anl.2021.05.007
  • Immune checkpoint inhibitors (ICIs) targeting the PD-1/PD-L1 axis induce sustained clinical responses in a sizable minority of cancer patients. A primary resistance to ICIs could be attributed to abnormal gut microbiome composition. An interesting paper reported antibiotics inhibited the clinical benefit of ICIs in patients with advanced cancer. Fecal microbiota transplantation (FMT) from cancer patients who responded to ICIs into germ-free or antibiotic-treated mice ameliorated the antitumor effects of PD-1 blockade, whereas FMT from nonresponding patients failed to do so. Metagenomics of patient stool samples at diagnosis revealed correlations between clinical responses to ICIs and the relative abundance of Akkermansia muciniphila Oral supplementation with A. muciniphila after FMT with nonresponder feces restored the efficacy of PD-1 blockade in an interleukin-12-dependent manner by increasing the recruitment of CCR9+CXCR3+CD4+ T lymphocytes into mouse tumor beds.  please cite doi:10.1126/science.aan3706
  • Although cancer-associated fibroblasts (CAFs) are crucial stromal cells, characterizing their heterogeneity is far from complete. A recent report analyzed a novel subset of CAFs in oral squamous cell carcinoma (OSCC), which positively expressed CD68, the classic marker of macrophages. The spatial and temporal distribution of the CD68+ CAF subset of OSCC (n = 104) was determined by CD68/actin alpha 2, smooth muscle (ACTA2+; α-SMA) immunohistochemistry of serial sections. The CD68+ α-SMA+ CAF subset was elevated from dysplasia to OSCC. Moreover, although both the tumor center and invasive front harbor an abundant CD68+ CAF subset, patients with low-CD68+ CAFs in the tumor center showed more recurrence after operation and shorter survival time, indicating the different function of CD68+ CAFs in tumor initiation and progression. Functional analysis in the OSCC-CAF co-culture system found knockdown of CD68 did not change the phenotype of CAFs, tumor growth, or migration. Unexpectedly, low-CD68+ CAFs were associated with aberrant immune balance. A high proportion of tumor-supportive Tregs was found in patients with low-CD68+ CAFs. Mechanistically, knockdown of CD68 in CAFs contributed to the up-regulation of chemokine CCL17 and CCL22 of tumor cells to enhance Treg recruitment. Thus, up-regulated CD68+ fibroblasts participate in tumor initiation, but the low-CD68+ CAF subset in OSCC is conducive to regulatory T-cell (Treg) recruitment in the tumor microenvironment and contribute to poor prognosis of OSCC patients. please cite doi:10.1016/j.ajpath.2019.12.007

Author Response

Point 1: Green typos in the text.

Response 1: The typos in the text has been improved.

Point 2: Some opinions to improve the overall quality:

Introduction

  1. the current surgical approach however also includes the use of innovative methods such as robotic surgery which can provide not only an approach in R0 but also in salvage surgery allowing to de-intensify adjuvant chemoradiotherapy, please cite doi:10.1016/j.anl.2021.05.007
  2. Immune checkpoint inhibitors (ICIs) targeting the PD-1/PD-L1 axis induce sustained clinical responses in a sizable minority of cancer patients. A primary resistance to ICIs could be attributed to abnormal gut microbiome composition. An interesting paper reported antibiotics inhibited the clinical benefit of ICIs in patients with advanced cancer. Fecal microbiota transplantation (FMT) from cancer patients who responded to ICIs into germ-free or antibiotic-treated mice ameliorated the antitumor effects of PD-1 blockade, whereas FMT from nonresponding patients failed to do so. Metagenomics of patient stool samples at diagnosis revealed correlations between clinical responses to ICIs and the relative abundance of Akkermansia muciniphila Oral supplementation with A. muciniphila after FMT with nonresponder feces restored the efficacy of PD-1 blockade in an interleukin-12-dependent manner by increasing the recruitment of CCR9+CXCR3+CD4+ T lymphocytes into mouse tumor beds. please cite doi:10.1126/science.aan3706
  3. Although cancer-associated fibroblasts (CAFs) are crucial stromal cells, characterizing their heterogeneity is far from complete. A recent report analyzed a novel subset of CAFs in oral squamous cell carcinoma (OSCC), which positively expressed CD68, the classic marker of macrophages. The spatial and temporal distribution of the CD68+ CAF subset of OSCC (n = 104) was determined by CD68/actin alpha 2, smooth muscle (ACTA2+; α-SMA) immunohistochemistry of serial sections. The CD68+ α-SMA+ CAF subset was elevated from dysplasia to OSCC. Moreover, although both the tumor center and invasive front harbor an abundant CD68+ CAF subset, patients with low-CD68+ CAFs in the tumor center showed more recurrence after operation and shorter survival time, indicating the different function of CD68+ CAFs in tumor initiation and progression. Functional analysis in the OSCC-CAF co-culture system found knockdown of CD68 did not change the phenotype of CAFs, tumor growth, or migration. Unexpectedly, low-CD68+ CAFs were associated with aberrant immune balance. A high proportion of tumor-supportive Tregs was found in patients with low-CD68+ CAFs. Mechanistically, knockdown of CD68 in CAFs contributed to the up-regulation of chemokine CCL17 and CCL22 of tumor cells to enhance Treg recruitment. Thus, up-regulated CD68+ fibroblasts participate in tumor initiation, but the low-CD68+ CAF subset in OSCC is conducive to regulatory T-cell (Treg) recruitment in the tumor microenvironment and contribute to poor prognosis of OSCC patients. please cite doi: 10.1016/j.ajpath.2019.12.007
  4.  

Response 2: Sentences and citations was added for all suggestions in the manuscript.

  1. Sentence and citation was added (Introduction):

Additionally, the current surgical approach has been further improved by the use of innovative methods such as robotic surgery, which allow not only an R0 approach but also a rescue operation that allows de-intensified adjuvant chemo radiotherapy [1].

  1. Sentence and citation was added (Introduction):

This primary resistance to ICIs could be due to an abnormal composition of the gut microbiome. Antibiotics have been shown to interfere with the clinical benefit of ICIs in patients with advanced cancer. Metagenomics of patients' stool samples at diagnosis revealed an association between clinical response to ICIs and the relative abundance of Akkermansia muciniphila. Oral supplementation with A. muciniphila after faecal microbiota transplantation with faeces from patients who did not respond to treatment increased the efficacy of PD-1 blockade in mouse tumours. Therefore, such a therapeutic approach could be useful for tumour patients [2]

  1. Sentence and citation was added (Introduction):

Although cancer-associated fibroblasts (CAFs) are important stromal cells, the characterisation of their heterogeneity is far from complete. It has recently been shown that up-regulated CD68+ fibroblasts are involved in tumour initiation, but the subset of CAFs with low CD68 expression in OSCC is conducive to the recruitment of regulatory T cells (Treg) in the tumour microenvironment and contributes to a poor prognosis of OSCC patients [3].

  1. Meccariello, G., et al., Neck dissection and trans oral robotic surgery for oropharyngeal squamous cell carcinoma. Auris Nasus Larynx, 2022. 49(1): p. 117-125.
  2. Routy, B., et al., Gut microbiome influences efficacy of PD-1-based immunotherapy against epithelial tumors. Science, 2018. 359(6371): p. 91-97.
  3. Zhao, X., et al., Diminished CD68(+) Cancer-Associated Fibroblast Subset Induces Regulatory T-Cell (Treg) Infiltration and Predicts Poor Prognosis of Oral Squamous Cell Carcinoma Patients. Am J Pathol, 2020. 190(4): p. 886-899.

Round 2

Reviewer 1 Report

All my comments are answered.